# The Impact of the Anticoagulant Type in Blood Collection Tubes on Circulating Extracellular Plasma MicroRNA Profiles Revealed by Small RNA Sequencing

**DOI:** 10.3390/ijms231810340

**Published:** 2022-09-07

**Authors:** Andrey V. Zhelankin, Liliia N. Iulmetova, Elena I. Sharova

**Affiliations:** Department of Molecular Biology and Genetics, Federal Research and Clinical Center of Physical-Chemical Medicine of Federal Medical Biological Agency, 119435 Moscow, Russia

**Keywords:** circulating microRNA, plasma, anticoagulant, miRNA-seq, extracellular biomarkers

## Abstract

Pre-analytical factors have a significant influence on circulating microRNA (miRNA) profiling. The aim of this study was a comprehensive assessment of the impact of the anticoagulant type in blood collection tubes on circulating plasma miRNA profiles using small RNA sequencing. Blood from ten healthy participants (five males and five females from 25 to 40 years old) was taken in collection tubes with four different anticoagulants: acid citrate dextrose (ACD-B), sodium citrate, citrate-theophylline-adenosine-dipyridamole (CTAD) and dipotassium-ethylenediaminetetraacetic acid (K2 EDTA). Platelet-free plasma samples were obtained by double centrifugation. EDTA plasma samples had elevated levels of hemolysis compared to samples obtained using other anticoagulants. Small RNA was extracted from plasma samples and small RNA sequencing was performed on the Illumina NextSeq 500 system. A total of 30 samples had been successfully sequenced starting from ~1 M reads mapped to miRNAs, allowing us to analyze their diversity and isoform content. The principal component analysis showed that the EDTA samples have distinct circulating plasma miRNA profiles compared to samples obtained using other anticoagulants. We selected 50 miRNA species that were differentially expressed between the sample groups based on the type of anticoagulant. We found that the EDTA samples had elevated levels of miRNAs which are abundant in red blood cells (RBC) and associated with hemolysis, while the levels of some platelet-specific miRNAs in these samples were lowered. The ratio between RBC-derived and platelet-derived miRNAs differed between the EDTA samples and other sample groups, which was validated by quantitative PCR. This study provides full plasma miRNA profiles of 10 healthy adults, compares them with previous studies and shows that the profile of circulating miRNAs in the EDTA plasma samples is altered primarily due to an increased level of hemolysis.

## 1. Introduction

Circulating extracellular microRNAs (miRNAs) have been widely studied as potential biomarkers of various human diseases [1]. Although the main biological role of miRNAs is realized inside the cell and consists of fine-tuning of gene expression by binding to mRNA. miRNAs are also released into the extracellular space and are involved in intercellular communication [2,3,4]. Circulating miRNAs (c-miRNAs) are stable in plasma and are protected from RNAse activity by binding to the circulating proteins or encapsulating within the extracellular vesicles: exosomes, microvesicles, or apoptotic bodies [5]. Of the known 2693 human mature miRNA species, several hundred are stably detected in the extracellular plasma fraction, forming specific profiles [6]. Typically, these profiles include a few species of highly abundant miRNAs, with the main diversity occurring only in a small fraction. The profiles of c-miRNAs are associated with different physiological and pathological conditions and may change with physical activity, injury and some types of drug therapy [7,8,9,10,11]. Changes in c-miRNA profiles associated with age, sex, smoking and even intraindividual differences have been described [12,13].

As the research into c-miRNAs as biomarkers of human pathologies has evolved, the data indicating the influence of pre-analytical factors on c-miRNA profiles has accumulated [14,15,16,17,18,19]. When studying extracellular c-miRNAs, cellular components must be removed and sample preparation conditions should be optimized to avoid cell degradation and release of cellular miRNAs [20]. This is primarily achieved by minimizing turbulence during blood collection and standardizing the storage time of the blood tube before obtaining the plasma. During plasma preparation, the removal of cellular components is usually achieved by centrifugation. Centrifugation conditions significantly affect c-miRNA profiles [21,22,23]. Two-step centrifugation with a repeated collection of the plasma supernatant has been shown to eliminate cellular components most effectively and is recommended for c-miRNA studies [18,21,24]. At the same time, the final fraction of plasma can vary in the number of residual platelets and microvesicles, depending on the centrifugation speed [22,25]. Platelets contain a variety of miRNAs, so it is important to specify centrifugation conditions and plasma type-platelet-rich plasma (PRP), platelet-poor plasma (PPP), or platelet-free plasma (PFP) in protocols for c-miRNA studies. It was shown that centrifugation at 60,000 g*min gets rid of more than 99% of platelets and obtained PFP [25].

Another important pre-analytical factor affecting c-miRNA profiles is the hemolysis caused by red blood cell (RBC) destruction [26,27]. RBCs contain large amounts of miRNAs that are released into the extracellular fraction as a result of hemolysis occurring in vivo or in vitro. Hemolysis should be controlled in every c-miRNA study and can be assessed visually based on the plasma sample coloration or spectrophotometrically by the free hemoglobin evaluation [28,29]. Changes in the levels of some hemolysis-dependent and hemolysis-independent miRNAs (e.g., miR-451a and miR-23a-3p) can additionally indicate the hemolysis level in plasma samples [28,30].

Plasma preparation conditions may also cause platelet activation and release of platelet microvesicles (PMVs) which contain specific miRNA signatures [31,32,33,34,35]. PMVs constitute the main fraction of all microvesicles in plasma [36,37,38]. Even when platelets are removed from plasma, PMV release during plasma preparation can significantly alter the initial profiles of c-miRNAs. Therefore, minimizing platelet activation in vitro is an important aspect of c-miRNA research. One of the pre-analytical factors affecting platelet activation and concentration of PMVs is the type of anticoagulant in blood collection tubes [39,40]. The influence of this factor on c-miRNA profiles has not yet been thoroughly investigated. Some studies have shown that both concentrations of activated platelets and PMVs are lower in citrate tubes than in EDTA tubes [41]. Anticoagulant and antiplatelet therapy affects circulating miRNA profiles, changing the plasma levels of platelet-derived miRNAs [31]. Mussbacher et al. have recently shown that the type of anticoagulant affects plasma levels of some miRNAs associated with platelet activation and a time-dependent increase in plasma miRNAs also depends on the anticoagulation type [42]. Significant correlations between miRNA levels and plasma concentrations of platelet-stored molecules pointed towards in vitro platelet activation.

In this study, we used small RNA sequencing to comprehensively assess the effect of the blood collection tube anticoagulant on healthy individual plasma c-miRNA profiles. By reducing the influence of the remaining pre-analytical factors to the required or technically possible minimum, we show the diversity of miRNAs and their differential expression using the several most common anticoagulants.

## 2. Results

### 2.1. Study Sample Characteristics

Ten healthy adults (five males and five females) were included in the study. The metadata of participants is provided in Table 1. From each participant, four plasma samples were obtained using different types of anticoagulants during blood collection (see the Materials and Methods section). The study groups were formed based on the type of anticoagulant:ACD: 10 samples obtained using acid citrate dextrose (ACD-B) blood tubes;Citrate: 10 samples obtained using sodium citrate 3.2% blood tubes;CTAD: 10 samples obtained using citrate-theophylline-adenosine-dipyridamole (CTAD) blood tubes;EDTA: 10 samples obtained using dipotassium-ethylenediaminetetraacetic acid (K2 EDTA) blood tubes.

Blood sampling was performed in the following order for all samples: sodium citrate, CTAD, ACD-B, K2 EDTA.

Characteristics of 40 biological samples, including the type of anticoagulant and the hemolysis score (HS) for plasma samples, miRNA concentration for miRNA samples and library preparation success for miRNA library samples, are provided in Table A1, Appendix B.

The results of a spectrophotometric assessment of hemolysis showed that the EDTA group had significantly elevated HS values compared to other groups (Kruskal-Wallis test, *p* < 0.05; Figure 1A). Clear peaks of free hemoglobin at the 414 nm wavelength were observed only in plasma samples obtained using K2 EDTA, while for other anticoagulants these peaks were absent or barely detectable. The concentration of miRNA did not differ significantly between the sample groups (Figure 1B).

### 2.2. miRNA Sequencing Statistics

Small RNA sequencing was performed for 36 of the 40 samples collected for the study excluding four samples with failed libraries. The total amount of miRNA reads for these samples after applying all filters was 80.1 M with a mean value of 2.2 ± 0.8 M. General data of miRNA sequencing (miRNA-seq) analysis is presented in Figure 2 and is additionally provided in Table A2, Appendix B.

Four samples (s03C, s05C, s09B and s10D) had a very low percentage of miRNA reads (<15%) and were excluded from the further analysis. Another two samples (s04D and s05A) had a lower percentage of miRNA reads and total number of detected miRNAs compared to other samples and their miRNA profiles did not correlate with profiles of other samples (Figure 2D). These two samples had a high ratio of unaligned reads (more than 50%) and were strong outliers on the PCA P1-P2 visualization plot, so they were also excluded from the further analysis. The final analysis included 30 samples with an average of 1.6 ± 0.5 M miRNA reads per sample (Figure 3). The mean percent of miRNA reads to total read number was 74.5 ± 14.3. The average number of miRNAs detected within one sample was 308 ± 35.

### 2.3. The Impact of Anticoagulation on Circulating miRNA Profiles

#### 2.3.1. The Diversity of miRNAs in Plasma

We found a total of 342 miRNAs with more than two counts per million (CPM) in at least 10 samples. The majority of miRNAs were detected in all four groups (317 species or 92.7%, Figure 4A). The groups of the citrate-based anticoagulants (ACD, Citrate and CTAD) had extended number of detected miRNAs and shared 13 miRNAs that were not detected in the EDTA group. We applied the alpha diversity approach to our data to help demonstrate differences in miRNA diversity between the compared groups. The Shannon index was chosen to estimate the alpha diversity of miRNAs within each sample. Alpha diversity of the EDTA group was significantly reduced (*p* < 0.05, Mann–Whitney U test) compared to other groups (Figure 4D).

The average number of detected miRNAs per sample reached 272 ± 4. It was slightly reduced in the CTAD group compared to other groups (Figure 4C). Top 20 miRNAs represented ~90% of total reads in EDTA samples, which was higher than in other groups (from 77 to 81%). The most abundant miRNAs in all samples were miR-451a and miR-486-5p, which both represented ~35–40% of reads in the ACD, Citrate and CTAD groups and ~65% of reads in the EDTA group (Figure 4D).

Since mature miRNAs are usually presented in various isoforms (isomiRs) resulting from their preprocessing, we analyzed the sequences of miRNA reads to identify the diversity of isoforms in the study sample according to the recent isomiR classification [43]. To filter out isoforms with very low prevalence, we analyzed only miRNA reads that met the following criteria: mean CPM of the corresponding miRNA >2 in at least 10 samples and mean read count of isoform >10. After filtering, 2990 isoforms remained for 302 miRNA species. On average, ~32% of miRNA reads matched exactly the canonical sequences of mature miRNAs from miRBase (Figure 5A). More than half of all miRNA sequences had modifications on the 3′-end, most of which were 3′-trimming (Figure 5A). Extensions on the 3′-end were present in ~20% of miRNA sequences, with a higher proportion of 3′-templated modifications (i.e., those in which the sequence matches the corresponding pre-miRNA) than 3′-non-templated modifications (Figure 5A). 5′-end modifications were present in ~10–15% of the sequences and their distribution by type was similar to that of the 3′-end modifications. Polymorphic modifications (i.e., those in which the miRNA sequence had single nucleotide polymorphisms compared to the canonical miRNA sequence) were present in ~5% of the sequences. The percentage distribution of all isoform types in the study samples (*N* = 30) is shown in Figure 5B. The sequences and counts of the most commonly represented isoforms that accounted for more than 10% of the expression of the corresponding miRNA are listed in Appendix A (615 isoforms for 300 miRNAs, including canonical sequences).

#### 2.3.2. miRNA Profiles by PCA

The samples formed two distinct clusters when visualizing the miRNA profiles on the PC1-PC2 plot (Figure 6A). One cluster contained mostly EDTA samples, while samples from this group were not present in the other cluster. The cluster containing EDTA samples consisted exclusively of samples with a high (more than 85%) percentage of miRNA reads (Figure 6B). The HS values of samples from this cluster were mostly higher than in other samples (Figure 6B).

#### 2.3.3. Differential Expression of miRNAs

We found 125 statistically significant pairwise comparisons for 50 DE miRNAs (Appendix A). All DE miRNAs had significant differences in expression in at least one pairwise comparison between the EDTA group and each of the other groups. Among the 31 miRNAs with elevated levels in the EDTA group, the most significant differences were observed for miR-16-5p, miR-486-5p, miR-484, miR-451a, miR-92a-3p, miR-16-2-3p and miR-25-3p (Figure 7). Most of these miRNAs are abundant in RBC and are associated with hemolysis. Among the 11 miRNAs with decreased levels in the EDTA group, the most significant differences were observed for miR-4433b-5p, miR-409-3p and miR-223-3p (Figure 7). Normalized CPM values by the study groups for each DE miRNA are shown in Appendix A.

#### 2.3.4. The Impact of Hemolysis and the Contribution of RBC-Derived and Platelet-Derived miRNAs

Since we found elevated levels of hemolysis in the EDTA samples and hemolysis-associated miRNAs were among the top DE miRNAs, we analyzed the effect of hemolysis on circulating plasma miRNA profiles. We took 20 miRNAs that were the most abundant in RBCs based on the most recent and comprehensive study on miRNA sequencing of erythrocytes of healthy adults [44]. Among them, we selected the top ten species based on their abundance in plasma to form the list of potential RBC-derived circulating extracellular miRNAs. Similarly, we selected the top 10 potential platelet-derived miRNA species based on previous studies revealing the presence of platelet-specific miRNAs in extracellular fraction or using small RNA sequencing or microarray screening to analyze platelet miRNome [31,32,33,38]. Platelet-derived circulating miRNA levels are an important marker of in vivo or in vitro platelet activation resulting in the release of PMVs which contain specific miRNA signatures [31,33,34,38]. By introducing the terms “RBC-derived miRNAs” and “platelet-derived miRNAs”, we are only indicating the potential origin of these miRNAs in plasma, while understanding that they are not RBC- or platelet-specific and have also been found in varying abundance in other types of cells or tissues.

Based on the previous study on the impact of hemolysis on the circulating miRNA profiles in plasma [45], we also selected miRNAs susceptible to hemolysis among the DE miRNAs. These miRNAs, apart from those already placed in the RBC-derived group, were marked as “Hemolysis-susceptible” (Figure 7). Interestingly, the plasma levels of RBC-specific miR-144-3p/-5p and miR-4732-3p, which are expressed almost exclusively in erythrocytes, are crucial to erythropoiesis and are located within the same gene cluster along with miR-451a [46], were among the list of DE miRNAs and were significantly (more than two-fold) elevated in EDTA samples. These miRNAs were also regarded as “Hemolysis-susceptible”.

DE miRNAs with elevated plasma levels in the EDTA samples form three main groups based on the log_2_CPM threshold set between 11 and 12 (white color in Figure 7). The first group represents the upper part in the Figure 7 and includes mostly RBC-derived miRNAs with high expression (log_2_CPM > 13) among all study groups. The second group includes miRNAs from miR-425-5p to miR-15b-5p in Figure 7 and represent mostly miRNAs susceptible to hemolysis, but not abundant in RBCs. The expression of these miRNAs in the EDTA group is still high (log_2_CPM > 11), while in other groups their expression is below the established threshold. The third group include miRNAs with low expression among all study groups.

We found that the proportion of RBC-derived miRNAs was higher in the EDTA group compared to the other groups, while the proportion of platelet-derived miRNAs was decreased (Figure 8A). At the same time, the ratio between miRNAs within the RBC-derived and platelet-derived groups did not change significantly (Figure 8B,C). One miRNA, namely miR-191-5p, appeared to belong to both RBC and platelet groups and we considered it as RBC-derived in the general analysis. The proportion of miRNAs susceptible to hemolysis, but not abundant in RBCs, was similar in the study groups (Figure 8A).

The majority of RBC-derived miRNAs (except for miR-181a-5p) were more abundant in EDTA samples compared to any other group of samples (Figure 9) and eight of them were among the list of DE miRNAs (Appendix A and Figure 7). The majority of platelet-derived miRNAs were less abundant in EDTA samples compared to any other group of samples (Figure 10) and two of them (miR-223-3p and miR-23a-3p) were among the list of DE miRNAs (Appendix A and Figure 7).

### 2.4. Validation of miRNA Sequencing Data by Quantitative PCR (qPCR)

Data obtained by qPCR is provided as quantitative cycle values (Cq) in Appendix A. All miRNAs were detectable in all 40 samples with the mean Cq values from 15.5 for the most abundant miR-451a to 27.3 for the lowly-expressed miR-30e-5p. Outlier samples with unconfident qPCR detection, for which the mean Cq value for all miRNAs exceeded 25, were excluded from further analysis (samples s03C, s04D, s05A, s05C and s10D). Interestingly, the same samples had a low ratio of miRNA reads to total sequencing reads and were excluded from the miRNA-seq data analysis. The final sample for qPCR data analysis included 35 samples with the following sample numbers in the study groups: ACD—nine samples; Citrate—ten samples; CTAD—eight samples; EDTA—eight samples.

First, we analyzed the distribution of the qPCR-based RBC-Platelet miRNA ratio and miRNA hemolysis ratio dCq(miR-23a-3p-miR-451a) in the study groups. Samples from the EDTA group had both elevated RBC-Platelet miRNA Ratio and dCq(miR-23a-3p-miR-451a) values compared to the other groups (Kruskal-Wallis test, *p* < 0.05; Figure 11). Notably, samples s10A and s04B, which clustered with the EDTA samples on the PCA plot and had a high percent of miRNA reads (Figure 6), also had high qPCR-based values of the RBC-Platelet miRNA ratio and dCq(miR-23a-3p-miR-451a), which indicates possible contamination of these samples by the RBC-derived miRNAs.

Secondly, we analyzed the distribution of qPCR-based miRNA relative plasma levels, normalized to the reference miRNA, miR-30e-5p. The distribution of the miRNA plasma relative levels in the sample groups is shown in Figure 12. We found that relative plasma levels of all analyzed RBC-derived miRNAs were elevated and levels of all analyzed platelet-related miRNAs were lowered in samples from the EDTA group compared to the other groups (Kruskal-Wallis test, *p* < 0.05).

To assess the concordance between the miRNA relative expression values calculated from the qPCR and sequencing data, we calculated relative miRNA expression levels normalized to miR-30e-5p based on the log_2_CPM values from miRNA-seq data (EXP_CPM_ values) and plotted these values versus qPCR-based miRNA plasma levels normalized to miR-30e-5p (Figure 13). The plots were built for each miRNA analyzed by qPCR for the 30 samples included in the miRNA-seq data analysis. Each plot indicated the presence of discrepancies between qPCR-based and sequencing-based miRNA expression levels. The best concordance between qPCR and sequencing data were found for miR-150-5p, miR-23a-3p and miR-451a, whereas for miR-126-3p and miR-21-5p the correspondence was the lowest. Both qPCR and sequencing data indicated an increase in the relative expression of RBC-derived miRNAs (miR-451a, miR-92a-3p and miR-16-5p) in the EDTA sample group.

Finally, we analyzed the effect of hemolysis on miRNA levels using combined spectrophotometry, miRNA-seq and qPCR data. Using the sequencing data, we calculated the difference between log_2_CPM values of miR-451a and miR-23a-3p to obtain the sequencing-based miRNA hemolysis ratio. HS and dCq(miR-23a-3p-miR-451a) were used as the spectrophotometry-based and qPCR-based values for hemolysis assessment, respectively. For these three values, we build the dot plots and calculated R-squared (*R*^2^) values for the linear regression (Figure 14). We found that spectrophotometric hemolysis assessment converges with the miR-451a and miR-23a-3p ratio based on both qPCR and sequencing data, but does not accurately reflect them (Spearman’s Rho 0.417, *p* = 0.013 and 0.513, *p* = 0.004, respectively). However, the ratio between these miRNAs showed good concordance between qPCR and sequencing data (Spearman’s Rho 0.873, *p* < 0.001).

## 3. Discussion

As this study was devoted to the comprehensive investigation of the one particular pre-analytical factor that affects c-miRNA profiles, we built the study design so that the influence of the other factors was standardized or reduced to the possible technical minimum. We used a plasma extraction protocol considering previous studies [21,25] to reduce the risk of hemolysis and to get rid of cellular components, including platelets, as much as possible to obtain PFP. In this study, we analyzed the whole circulating extracellular miRNA pool, which includes exosome- and MV-derived miRNAs and miRNA-protein complexes.

We chose miRNA extraction and miRNA sequencing strategies based on the previously published data on the successful circulating miRNA sequencing case obtaining more than 60% of miRNA reads [47]. In our case, four samples failed to obtain miRNA libraries. The cause of this phenomenon cannot be identified, since these samples did not differ significantly from the others either in miRNA concentration or in Cq values obtained by qPCR. These samples likely failed for random reasons that occurred during the sample preparation workflow since the construction of miRNA libraries has many separate steps and is challenging for miRNA samples extracted from biofluids. At the initial stage of the study, we analyzed several miRNA samples using a Bioanalyzer 2100 instrument with a Small RNA Kit (Agilent, Santa-Clara, CA, USA) to estimate the presence of miRNA fraction. However, no peaks were detected on the electrophoregram due to low RNA concentration, so this technique was not further used in the study. The Qubit miRNA assay measured the miRNA concentration at the lower limit of detection, so we assume that the results of this measurement may be inaccurate. Given the small amounts of miRNA per input, we chose to use a maximum volume of sample input (9.5 µL), a maximum number of amplification cycles (25 cycles) on the PCR step and a PAGE option on the size selection step during the miRNA library preparation.

Using this combination of plasma isolation, miRNA extraction and miRNA library preparation techniques, we obtained about 1 M miRNA sequencing reads for 30 of the 36 samples. The exclusion of six samples with a low percentage of miRNA reads was justified because their presence in the sample would have led to bias due to their low miRNA diversity. Within the 30 samples included in the analysis, miRNAs represented the major fraction of small RNAs. Although we used YRNA/tRNA Blockers during the small RNA library preparation, from 0.7 to 20.2% of sequencing reads mapped to YRNAs. These reads mostly represented short 5′-end YRNA fragments with length similar to miRNAs. Reads mapped to YRNA were most common after miRNAs within the small RNA fraction, representing 50 to 95% of the residual small RNA reads. The presence of this fraction within the circulating small RNA pool has been described previously [48,49].

To estimate the relevance of the miRNA-seq data obtained in our study, we compared our results with the results of the previous study by Wong et al., 2019 [47] which used the same protocol for miRNA extraction and sequencing to obtain plasma miRNA profiles. The mean percentage of miRNA reads and the number of detected miRNA species per sample were similar between the studies (~60–70% and ~300–350 species, respectively) and the lists of the top 20 miRNA species overlapped by 70%. IsomiR composition in the analyzed samples was consistent with previously published data on the general isomiR distribution [43]. The percentage of canonical miRNAs as long as 3′ and 5′ miRNA modifications were similar to those reported in a previous study analyzing isomiRs in plasma miRNA pool [50].

We also compared the miRNA-seq data from this study with the previous studies using small RNA sequencing to explore plasma c-miRNA profiles in the groups of healthy adults. We analyzed the overlapping between the top 20 miRNA species based on the mean CPM values in this study and in studies by Godoy et al., 2018 [49] and Max et al., 2018 [51] (Figure 15). Among the 37 unique miRNAs from all lists, eight species were common in all three studies. Our study shared 11 miRNAs with each of the analyzed studies. Six miRNA species from the top 20 plasma miRNAs in our study were not observed in the top 20 miRNA lists from the two analyzed studies. The differences between the studies are probably due to different protocols for plasma preparation, miRNA extraction and sequencing technology. The results of this study complement existing data on plasma miRNA sequencing in healthy adults and contribute to the concept of a “normal” c-miRNA profile.

The main finding of this study is the difference in c-miRNA profiles between plasma samples obtained using K2 EDTA blood collection tubes and plasma samples obtained using other anticoagulants—ACD-B, sodium citrate and CTAD. We found that this difference is primarily due to a shift in the ratio between RBC-derived and platelet-derived miRNAs, which most likely occurred as a result of an elevated level of RBC hemolysis in the EDTA samples.

The issue of elevated hemolysis in the EDTA tubes has not previously been described. Mussbacher et al. [42] showed a significant increase in miR-451a plasma level in plasma samples obtained using EDTA compared to samples obtained using sodium citrate and CTAD. This observation indicates a potential increase of hemolysis in the EDTA samples.

We observed that the percentage of miRNA reads in small RNA sequencing data was higher in samples obtained using EDTA tubes compared to samples obtained using other anticoagulants. However, in the EDTA sample group, three highly abundant miRNAs related to hemolysis (miR-451a, miR-486-5p and miR-92a-3p) represented more than 70% of all miRNA reads, while in other sample groups they represented from 40 to 50% of miRNA reads. The alpha diversity approach showed that the diversity of the detected miRNAs was lower in the EDTA samples compared to the other sample groups. The choice of K2 EDTA as an anticoagulant for blood collection for plasma miRNA-seq studies might increase the chances of successful library preparation, but it results in reduced miRNA diversity and biased miRNA profiles due to hemolysis.

Citrate-based anticoagulants including ACD, sodium citrate and CTAD are mainly used for the preparation of PRP, which can be stored in biobanks or used in the studies involving platelet isolation, performing platelet function tests and determining plasma concentrations of platelet-stored molecules. The use of these anticoagulants reduces the level of in vitro platelet activation, which is also important for the studies on plasma c-miRNA profiles [31,33,34]. The study by Mussbacher et al. revealed the increased plasma levels of several platelet-derived miRNAs (miR-191-5p, miR-320a, miR-21-5p and miR-23a-3p) in the EDTA plasma compared to CTAD and citrate plasma [42]. In our study, levels of miR-191-5p and miR-320a were only slightly increased in the EDTA samples compared to other sample groups, whether the majority of platelet-derived miRNAs had lowered levels in the EDTA samples. As mentioned before, miR-191-5p is highly represented both in platelets and in RBCs and its levels may be elevated due to hemolysis.

In this study, we found that the choice of anticoagulant for blood collection significantly influences the ratio between the platelet-derived and RBC-derived miRNA in plasma, with the highest percentage of RBC-derived miRNAs (80%) in the EDTA group and the lowest percentage (45%) in the CTAD group. The decreased levels of most platelet-derived miRNAs in the EDTA group in our study are most likely the result of this bias rather than indicating reduced platelet activation. We also found that the plasma levels of some miRNAs that are susceptible to hemolysis based on the previous studies, but are not abundant in RBCs, are elevated in the EDTA group compared to other sample groups.

The change in the ratio of RBC-derived and platelet-derived miRNAs was the main criterion for the selection of miRNA targets to validate the sequencing data by qPCR. We found discrepancies in the expression levels of some miRNAs between the qPCR and sequencing data, which may occur due to differences in the cDNA synthesis and amplification efficiency, ligation bias, or the influence of isomiR variance [52,53]. Despite these discrepancies, the qPCR data confirmed the sequencing data with respect to the shift between RBC-derived and platelet-derived miRNAs in the EDTA group compared to other sample groups.

The main limitation of this study is the loss of single samples due to the low quality of miRNA libraries, which shifted the composition of the compared groups and, from the originally planned 40 samples, only 30 were included in the analysis. This problem could not be solved because most of the miRNA sample volume was used to prepare libraries without the possibility of reuse.

Another limitation is the lack of measurement of the total blood count and its biochemical parameters, which could be used to estimate changes in the parameters related to the blood cellular components. In addition, platelet count, PMV count and platelet activation were not measured in plasma samples included in this study. We state that we obtained PFP only on the basis of previous studies, but we did not measure the amount of the residual platelets in plasma.

The study was limited by the number of miRNAs included in the qPCR analysis. This analysis was performed only to assess the RBC-Platelet miRNA ratio and not all DE miRNAs were validated by qPCR.

## 4. Materials and Methods

### 4.1. Study Sample

Blood collection was performed from Caucasian healthy individuals from 25 to 40 years old without obesity or underweight (BMI from 18 to 30). The study was approved by the local ethics committee and written informed consent was obtained from each study participant. The exclusion criteria were the following: smoking; taking anticoagulants, antiplatelet drugs, ACE inhibitors, calcium antagonists, beta-blockers, or statins in the last month; any active malignancies; diabetes; cardiovascular diseases; pulmonary, renal, or hepatic insufficiency; acute infections in the last week; any surgical operations in the last week; menstruation; pregnancy; lactation; past-week history of gastrointestinal symptoms (diarrhea, vomiting); alcohol and drug abuse; inability or unwillingness to provide written consent. Fasting blood samples were collected from each participant in VACUETTE vacuum blood collection tubes with four different types of anticoagulants: ACD-B, sodium citrate 3.2%, CTAD and K2 EDTA (Greiner Bio-One, Kremsmünster, Austria). The volume of blood for each tube varied from 3.5 to 4 ml. After blood collection, tubes were gently inverted ten times and stored at room temperature prior to plasma isolation.

### 4.2. Plasma Preparation

Within 30 min after blood collection, blood tubes were centrifuged at 1000× *g* for 10 min at 21 °C in a swinging-bucket rotor centrifuge Eppendorf 5702R (Eppendorf, Hamburg, Germany). Plasma supernatant from each tube was carefully transferred into a new sterile 15 mL conical tube, leaving ~0.5 cm of plasma above the buffy coat layer. Conical tubes were centrifuged again in the same centrifuge at 2500× *g* for 20 min at 21 °C. General conditions of centrifugation speed and time reached 60,000× g*min, which allows removal of 99% of platelets and obtaining PFP [25]. The upper two-thirds of PFP was collected and stored at −80 °C in 600 μL portions in sterile RNAse-free 1.5 mL tubes. An additional 20 μL of PFP from each sample was transferred to a 200 μL PCR tube and was immediately used for hemolysis assessment.

### 4.3. Hemolysis Assessment of Plasma Samples

Only samples without visually detected hemolysis were included in this study. Low-level RBC hemolysis in plasma samples was assessed via the spectrophotometric measurement of absorbance at the 414 nm wavelength (peak of free hemoglobin). Plasma samples were analyzed on NanoDrop^®^ 2000 spectrophotometer (Thermo Fisher Scientific, Waltham, MA, USA) via the measurement of ultraviolet-visible (UV-vis) absorbance with a 1 mm path at 385 nm (A385) and 414 nm (A414) wavelengths. For each measurement, a lipemia-independent hemolysis score was calculated based on the A414 and A385 values: HS = ∆(A414 − A385) + 0.16 ∗ A385 [29].

### 4.4. Plasma miRNA Isolation

Isolation of miRNA was performed in batches including each four plasma samples obtained with different types of anticoagulants from the same individual. Frozen plasma samples were placed at room temperature for 5 min and then at 37 °C for 1–2 min until completely thawed. Samples were gently vortexed and miRNA was isolated from 600 μL of plasma using a NextPrep Magnazol cfRNA Isolation Kit (PerkinElmer, Waltham, MA, USA) according to the manufacturer’s guidelines. Each miRNA sample had a total volume of 16 µL and was stored at −80 °C prior to miRNA library preparation. The concentration of miRNA was measured using a Qubit miRNA Assay (Thermo Fisher Scientific Inc., Waltham, MA, USA) and a Qubit 3.0 fluorimeter (Thermo Fisher Scientific, Waltham, MA, USA).

### 4.5. miRNA Sequencing

Libraries for miRNA sequencing were prepared using a NEXTFLEX Small RNA-Seq Kit v3 (PerkinElmer, Waltham, MA, USA) according to the manufacturer’s guidelines. The input volume was 9.5 µL for each miRNA sample. NEXTFLEX tRNA/YRNA Blockers (PerkinElmer, Waltham, MA, USA) were added to each sample on the first step of library preparation to deplete tRNAs and YRNAs in a small RNA pool. PCR amplification included 25 cycles. PAGE size selection and the cleanup protocol were used to obtain fraction of miRNA libraries. The final volume of each library was 12 µL. The size distribution of miRNA libraries was assessed by a High Sensitivity DNA Assay on a Bioanalyzer 2100 capillary electrophoresis system (Agilent Technologies, Santa Clara, CA, USA). The concentration of miRNA libraries was measured by a Qubit dsDNA HS Assay and a Qubit 3.0 fluorimeter (Thermo Fisher Scientific, Waltham, MA, USA). The size range of the libraries was 150–160 nucleotides. Successful library preparation (i.e., the presence of a visible miRNA library band on PAGE and the presence of a peak of appropriate size on the electrophoregram) was realized for 36 of 40 samples. The size range of the purified libraries was 150–160 nucleotides. The libraries were pooled in an equimolar ratio. The size distribution of the library pool on a Bioanalyzer 2100 electrophoregram is shown in Appendix A. Sequencing of miRNA library pool was performed on a NextSeq 500 System using the NextSeq 500/550 High Output Kit v2.5 (Illumina, San-Diego, CA, USA) in a single-end mode with 75 cycles.

### 4.6. Sequencing Data Analysis

A total of 121.7M sequencing reads were obtained for all samples. Sequencing reads quality was analyzed using FastQC. Sequence reads were trimmed for the 3′ adapter sequence (TGGAATTCTCGGGTGCCAAGG) using Cutadapt. 4N barcodes were then removed by trimming the first and last 4 bases from each read. Reads with lengths less than 16 and more than 28 were removed. After trimming and filtering, the total number of reads was reduced to 84.9M. The absolute majority of reads had high quality scores (Appendix A).

QuickMIRSeq pipeline was used for miRNA-seq analysis [54]. Sequence reads were mapped to miRBase version 21 to obtain read counts and RPM values for each miRNA. An optional “Remapping” step was used to map miRNA sequences with mismatches to the reference human genome to reduce the number of likely false positives. Potentially noisy reads were filtered out by removing reads with an average number per sample of 2 or less and missing in at least 90% of samples. MiRNAs with more than 2 counts per million (CPM) in at least 10 samples were selected for all downstream analyzes.

The alpha diversity approach was applied to estimate the miRNA diversity in the study groups. The Shannon index is an abundance-sensitive measure of species diversity within a community and is commonly used as an alpha diversity metric in microbial high-throughput sequencing analysis and population genetics [55,56]. The Shannon indices were calculated based on the CPM values using the vegan R package (v. 2.6.2) implementation. Differences in the alpha diversity were determined by means of Mann-Whitney U-test performed on each pair of groups.

IsomiR analysis was performed based on the QuickMIRSeq data. Principal component analysis (PCA) was performed by the PCAtools R package v. 2.2.0 using scaled log_2_CPM values.

Differential expression analysis was conducted using EdgeR v. 3.32.1 [57]. The method of the trimmed mean of M-values (TMM) was applied for the normalization of the library sizes. Differentially expressed (DE) miRNAs between groups corresponding to anticoagulant types were identified using the pairwise quasi-likelihood F-test to find differences in expression between each pair of compared groups. Since each individual was present in at least two comparison groups, we adjusted the model for baseline differences between the individuals by assigning subject ID as a blocking variable. To control the false discovery rate, we adjusted *p*-values using the Benjamini-Hochberg method across all contrasts simultaneously. Comparisons with adjusted *p*-value < 0.05 were considered statistically significant. Of all miRNAs with log_2_CPM > 5 which had statistically significant comparisons, only those with absolute log_2_FC values > 1 or having at least three significant comparisons were considered as DE miRNAs.

### 4.7. Validation of miRNA Expression by qPCR

A 2 µL sample of miRNA was used for cDNA synthesis with a TaqMan Advanced miRNA cDNA Synthesis Kit (Thermo Fisher Scientific, Waltham, MA, USA), according to the manufacturer’s recommendations, using a DNA Engine Tetrad 2 Thermal Cycler (Bio-Rad Laboratories, Hercules, CA, USA). Eleven commercially available TaqMan Advanced miRNA assays with TaqMan Fast Advanced Master Mix (Thermo Fisher Scientific, Waltham, MA, USA) were used to perform qPCR, according to the manufacturer’s protocol. The list of miRNA assays with catalog numbers and mature miRNA sequences is given in Table 2. The list includes four miRNAs that are abundant in platelets or PMVs and three miRNAs abundant in RBCs and linked with RBC hemolysis. For qPCR-based hemolysis assessment, we used a pair of miRNAs (hsa-miR-23a-3p and hsa-miR-451a) that indicate the degree of hemolysis by their ratio [28,30]. Reference miRNA for normalization of the qPCR data, miR-30e-5p, was chosen on the basis of the previous study which analyzed the stability of miRNAs in plasma [58]. The stability of miRNA expression in plasma was assessed using the NormFinder algorithm [59] on the qPCR data and miR-30e-5p showed the lowest stability value, indicating that it was the most stable miRNA. The stability values are provided in Table A3, Appendix B. For each miRNA assay, a no-template control (NTC) containing nuclease-free water instead of a miRNA sample was analyzed. We performed qPCR by using the QuantStudio 5 Real-Time PCR system (Thermo Fisher Scientific, Waltham, MA, USA) in MicroAmp 96-well PCR plates and optical adhesive film, in a “Fast” cycling mode with the following program: enzyme activation—20 s at 95 °C; 45 cycles, denature—1 s at 95 °C, anneal/extend—20 s at 60 °C. We obtained qPCR data by using QuantStudio Design and Analysis Software v1.4.1 (Thermo Fisher Scientific, Waltham, MA, USA). Cq values were calculated by using the automatic “Baseline” value and the experimentally set “Threshold” value of ∆Rn = 0.15 for all analyzed miRNA targets. Cq measurements were performed in a single technical replicate for each miRNA target within an individual sample. For each sample, the ratio between RBC-related and platelet-related miRNAs (RBC-Platelet miRNA ratio) was calculated as an absolute value of the difference between the mean Cq of RBC-related miRNAs (miR-451a, miR-92a-3p and miR-16-5p) and the mean Cq of platelet-related miRNAs (miR-223-3p, miR-126-3p, miR-21-5p and miR-150-5p). The impact of RBC hemolysis on circulating miRNAs was also estimated based on the qPCR-detected ratio of miR-451a and miR-23a-3p. The difference between Cq values of these miRNAs was calculated by the formula: dCq(miR-23a-3p–miR-451a) = Cq(hsa-miR-23a-3p)–Cq(hsa-miR-451a). Relative plasma levels for each miRNA were calculated as expression levels normalized to the reference miRNA miR-30e-5p by the formula: EXP_miRNA = Cq(miR-30e-5p)—Cq(miRNA). For all the samples in this study, the same laboratory workflow and data analysis protocol was used.

## 5. Conclusions

Using miRNA sequencing, we showed that the choice of anticoagulant in blood collection tubes is the pre-analytical factor that affects plasma c-miRNA profiles. We found that using EDTA blood collection tubes results in elevated hemolysis in plasma samples compared to other anticoagulants. This leads to a shift in plasma c-miRNA profiles, primarily by changing the ratio between platelet-derived and RBC-derived miRNAs, which was validated by qPCR. At the same time, we found no significant differences in plasma c-miRNA profiles between the citrate-based tubes (ACD, sodium citrate and CTAD). We obtained full plasma miRNA profiles for ten healthy adults and compared them with previous studies, contributing to the concept of a “normal” c-miRNA profile.

## Figures and Tables

**Figure 1 ijms-23-10340-f001:**
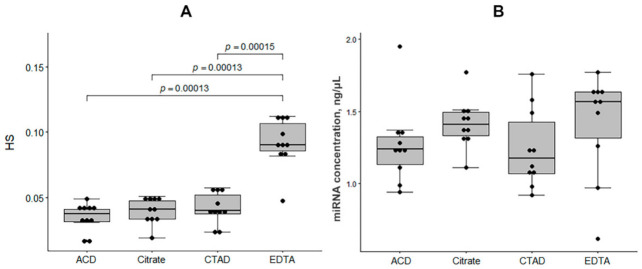
The distribution of biological sample characteristics in the study sample groups (*N* = 40). (**A**) Hemolysis score (HS). (**B**) Concentration of miRNA, ng/μL. The boxplots represent median and interquartile ranges (IQRs) in the box and values for individual samples in the dots. The Mann-Whitney U test *p*-values corrected by false discovery rate are given for the statistically significant pairwise comparisons.

**Figure 2 ijms-23-10340-f002:**
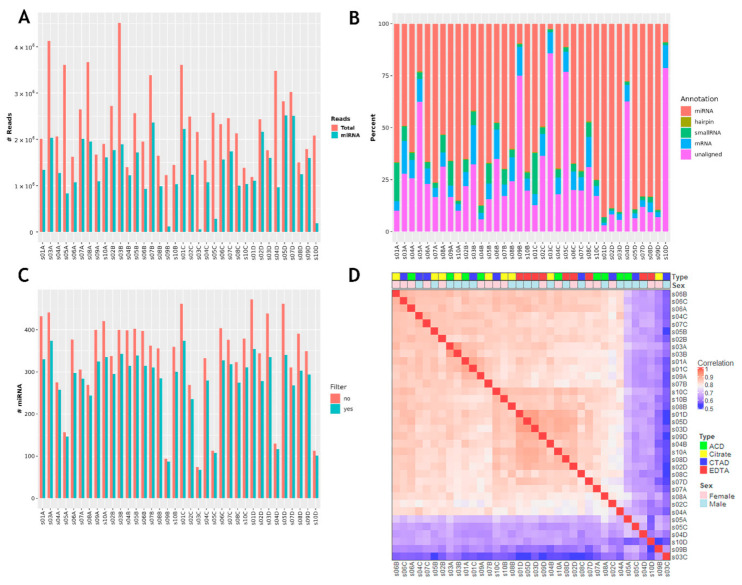
General output of miRNA sequencing analysis using the QuickMIRSeq pipeline. (**A**) Total and miRNA reads in each sample. (**B**) Distribution of miRNA reads compared to other types of reads. (**C**) Number of detected miRNAs per sample with and without filtering of noisy reads. (**D**) Correlation of miRNA expression between samples.

**Figure 3 ijms-23-10340-f003:**
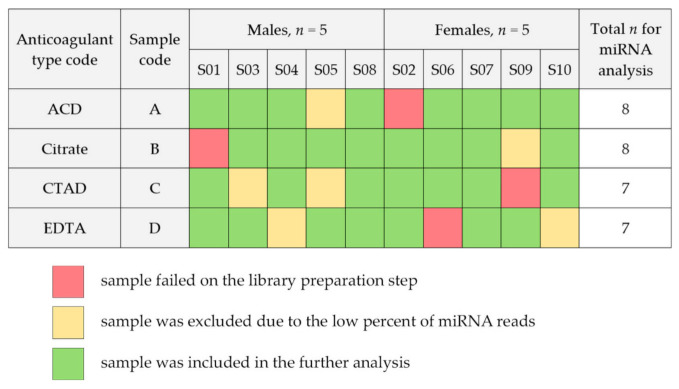
The study samples included in the further analysis.

**Figure 4 ijms-23-10340-f004:**
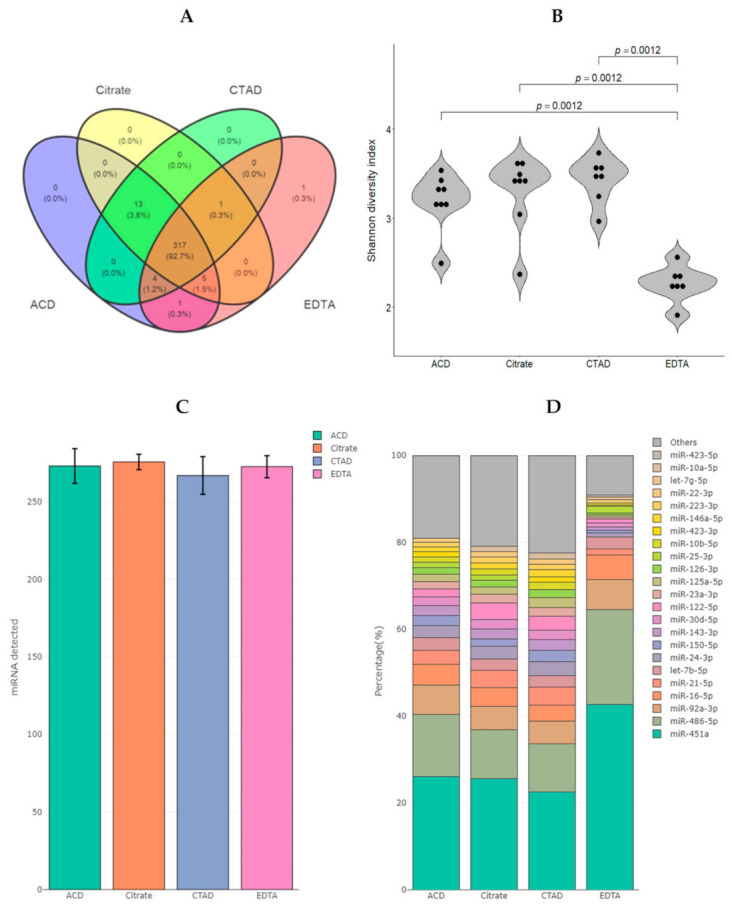
Diversity of miRNAs in plasma samples (*N* = 30) obtained using different anticoagulants in blood collection tubes. (**A**) Venn diagram for the total numbers of detected miRNAs with more than 2 counts per million (CPM) in at least 10 samples in the study groups. (**B**) Grouped violin plots reflecting the distribution of the alpha diversity represented by Shannon indices. The dots correspond to individual samples. The Mann-Whitney U test *p*-values corrected by false discovery rate are given for the statistically significant pairwise comparisons. (**C**) The average number of detected miRNAs per sample in the study groups. Error bars indicate standard error of the mean. (**D**) Representation of the top 20 miRNAs in each study group by CPM values.

**Figure 5 ijms-23-10340-f005:**
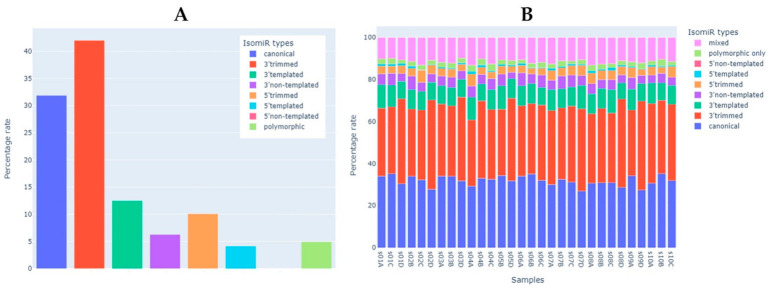
The analysis of miRNA isoforms (isomiRs). (**A**) Presence of different isomiR types in the study sample (*N* = 30). (**B**) Percentage of each isomiR type in each study sample.

**Figure 6 ijms-23-10340-f006:**
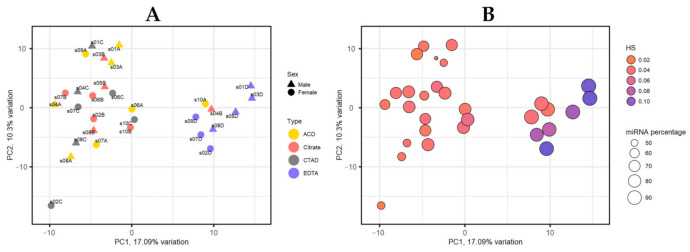
Principal component analysis of miRNA profiles by the PCAtools R package, version 2.2.0. (**A**) PC1-PC2 plot showing the ID and sex of participants and study groups based on the anticoagulant type. (**B**) PC1-PC2 plot showing hemolysis score (HS) values and percentage of miRNA reads.

**Figure 7 ijms-23-10340-f007:**
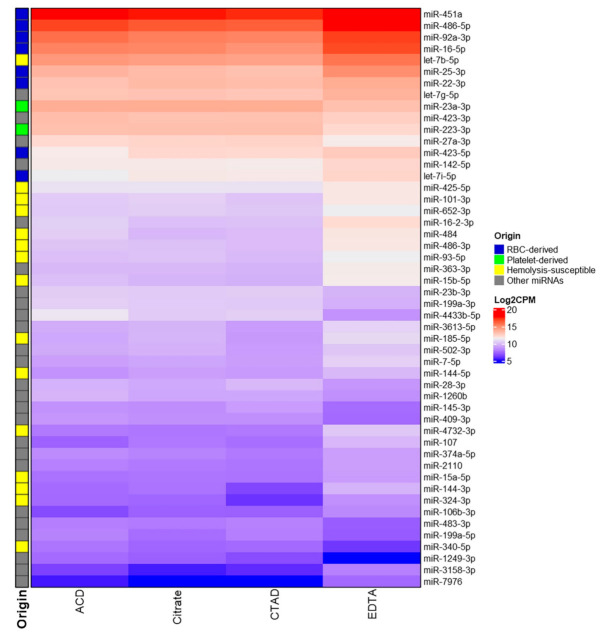
The heatmap of the log_2_CPM values for differentially expressed miRNAs in the study groups. The potential origin of miRNAs corresponds to the terms “RBC-derived miRNAs” and “Platelet-derived miRNAs” based on the top 10 miRNAs in each cell type (see Section 2.3.4), “Hemolysis-susceptible miRNAs” based on the additional miRNAs linked with hemolysis based on the previous studies, or indicate other miRNAs not belonging to these groups (“Other miRNAs”).

**Figure 8 ijms-23-10340-f008:**

The percentage of RBC-derived, platelet-derived and hemolysis-susceptible miRNAs in the study groups. (**A**) Distribution among all miRNA reads. (**B**) Distribution among the top 10 RBC-derived miRNAs. (**C**) Distribution among the top 10 platelet-derived miRNAs. The percentage was calculated based on the TMM-normalized CPM values.

**Figure 9 ijms-23-10340-f009:**
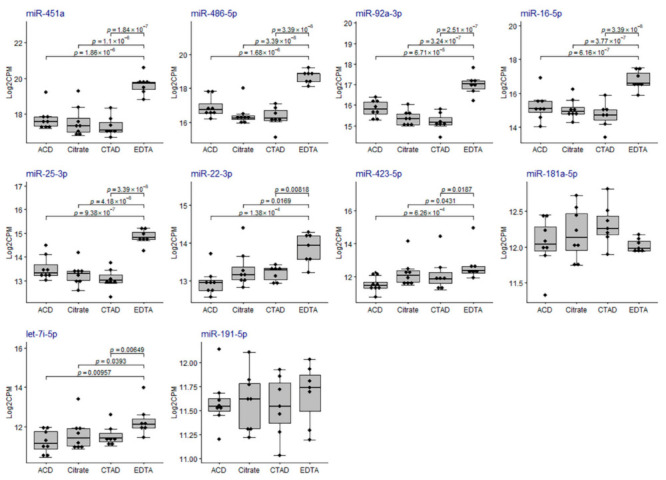
The distribution of miRNA expression for the top 10 RBC-derived miRNAs in the study sample groups. On the *y*-axis, TMM-normalized log_2_CPM values are shown. The boxplots represent median and interquartile ranges (IQRs) in the box and values for individual samples in the dots. *p*-values of statistical significance for the pairwise quasi-likelihood F-test comparisons performed in EdgeR are provided on the plots for each pair of the study groups with *p* < 0.05.

**Figure 10 ijms-23-10340-f010:**
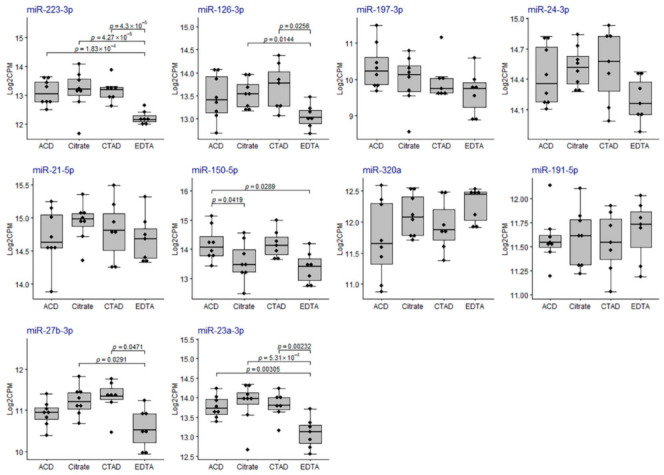
The distribution of miRNA expression for the top ten platelet-derived miRNAs in the study sample groups. On the *y*-axis, TMM-normalized log_2_CPM values are shown. The boxplots represent median and interquartile ranges (IQRs) in the box and values for individual samples in the dots. *p*-values of statistical significance for the pairwise quasi-likelihood F-test comparisons performed in EdgeR are provided on the plots for each pair of the study groups with *p* < 0.05.

**Figure 11 ijms-23-10340-f011:**
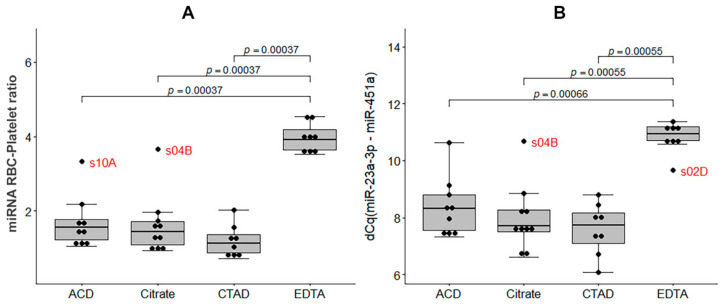
(**A**) The distribution of the qPCR-based RBC-Platelet miRNA ratio in the study sample groups. (**B**) The distribution of the qPCR-based miRNA hemolysis ratio dCq(miR-23a-3p-miR-451a) in the study sample groups. The boxplots represent median and interquartile ranges (IQRs) in the box and values for individual samples in the dots. The Mann–Whitney U test *p*-values corrected by false discovery rate are given for the statistically significant pairwise comparisons.

**Figure 12 ijms-23-10340-f012:**
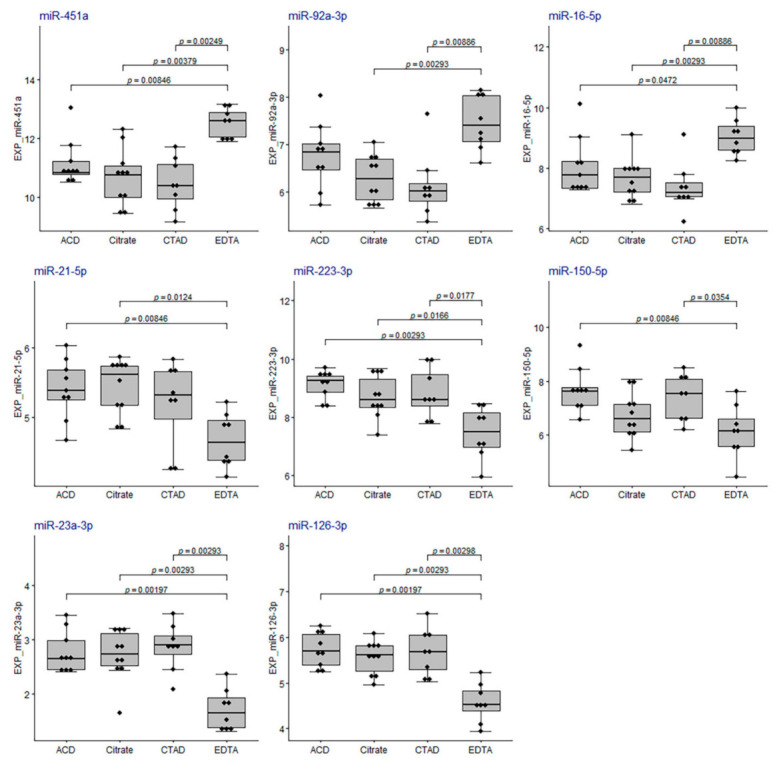
The distribution of the relative plasma levels of miRNAs normalized to miR-30e-5p. On the *y*-axis, relative plasma levels are marked as EXP_miR-… for each analyzed miRNA. The boxplots represent median and interquartile ranges (IQRs) in the box and values for individual samples in the dots. The Mann–Whitney U test *p*-values corrected by false discovery rate are given for the statistically significant pairwise comparisons.

**Figure 13 ijms-23-10340-f013:**
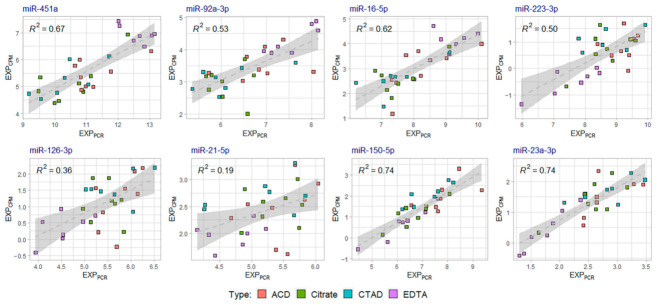
The concordance between qPCR-based and sequencing-based miRNA relative expression values within the 30 samples included in the miRNA sequencing analysis: mean EXP_CPM_ values vs. mean EXP_PCR_ values plot. EXP_CPM_ values were calculated based on miRNA sequencing data as the difference between the mean log_2_CPM value of the analyzed miRNA and the mean log_2_CPM value of miR-30e-5p. EXP_PCR_ values were calculated as qPCR-based miRNA plasma levels normalized to miR-30e-5p. The dots represent individual samples included in the analysis, the dotted line represents a trend line for a linear regression and the gray areas correspond to the 95% confidence interval. For each plot, R-squared (*R*^2^) values for the linear regression are given. *R*^2^ values indicate the proportion of the variance for a dependent variable (*y*-axis) that is explained by an independent variable (*x*-axis).

**Figure 14 ijms-23-10340-f014:**
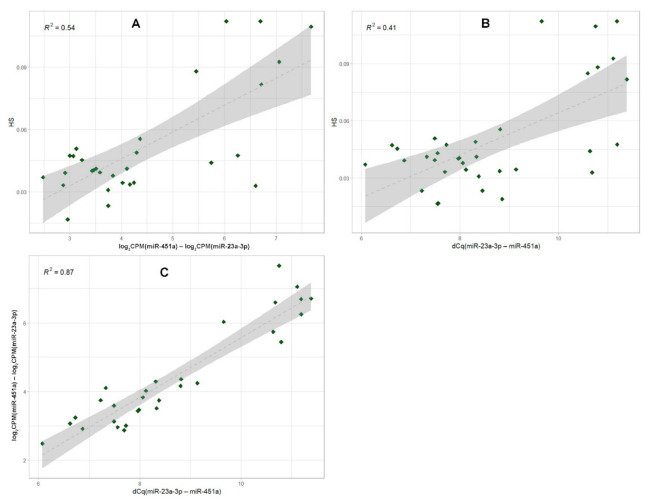
Correlation plots for the hemolysis assessment using different methods. (**A**) Spectrophotometric hemolysis score (HS) vs. sequencing-based miRNA hemolysis ratio between miR-23a-3p and miR-451a based on log_2_CPM values. (**B**) HS vs. qPCR-based miRNA hemolysis ratio dCq(miR-23a-3p-miR-451a). (**C**) Sequencing-based miRNA hemolysis ratio vs. qPCR-based miRNA hemolysis ratio dCq(miR-23a-3p-miR-451a). The dots represent the individual samples, the dashed line represents a linear regression and the gray areas correspond to the 95% confidence interval. For each plot, R-squared (*R*^2^) values for the linear regression are given. *R*^2^ values indicate the proportion of the variance for a dependent variable (*y*-axis) that is explained by an independent variable (*x*-axis).

**Figure 15 ijms-23-10340-f015:**
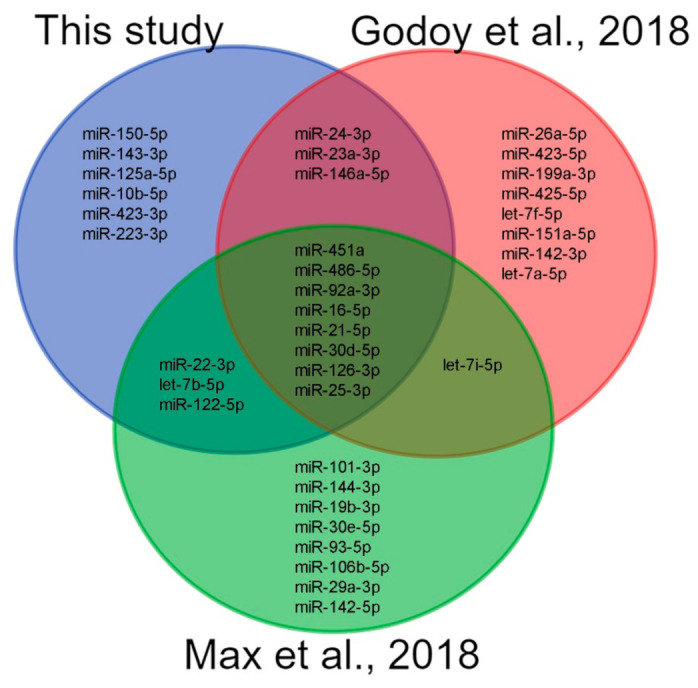
Venn diagram showing the overlapping between the top 20 miRNA species lists from this study and two previous studies using small RNA sequencing to explore circulating plasma miRNA profiles in the groups of healthy adults. This study: 30 samples from 10 individuals; Godoy et al. [49]: 12 samples from 12 individuals; Max et al. [51]: 159 samples with more than 0.5M miRNA reads from 13 individuals.

**Table 1 ijms-23-10340-t001:** The metadata of the study participants.

Individual ID	Sex	Age, years	BMI, kg/m^2^
s01	male	31.2	28.3
s02	female	26.8	20.8
s03	male	28.7	20.8
s04	male	32.5	27.4
s05	male	34.7	22.3
s06	female	34.9	22.2
s07	female	34.1	20.4
s08	male	33.2	20.3
s09	female	29.6	21.8
s10	female	37.7	18.7

**Table 2 ijms-23-10340-t002:** List of the miRNA assays used for qPCR.

Assay Name	Assay ID	Mature miRNA Sequence	Type of miRNA
hsa-miR-223-3p	477983_mir	UGUCAGUUUGUCAAAUACCCCA	Platelet-derived
hsa-miR-126-3p	477887_mir	UCGUACCGUGAGUAAUAAUGCG	Platelet-derived
hsa-miR-21-5p	477975_mir	UAGCUUAUCAGACUGAUGUUGA	Platelet-derived
hsa-miR-150-5p	477918_mir	UCUCCCAACCCUUGUACCAGUG	Platelet-derived
hsa-miR-16-5p	477860_mir	UAGCAGCACGUAAAUAUUGGCG	RBC-derived
hsa-miR-92a-3p	477827_mir	UAUUGCACUUGUCCCGGCCUGU	RBC-derived
hsa-miR-451a	478107_mir	AAACCGUUACCAUUACUGAGUU	RBC-derived/Hemolysis assessment
hsa-miR-23a-3p	478532_mir	AUCACAUUGCCAGGGAUUUCC	Hemolysis assessment
hsa-miR-30e-5p	479235_mir	UGUAAACAUCCUUGACUGGAAG	Normalization control

## Data Availability

The sequencing data are available in the Gene Expression Omnibus repository (GSE200970).

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
