# Peer review of "The Impact of the Anticoagulant Type in Blood Collection Tubes on Circulating Extracellular Plasma MicroRNA Profiles Revealed by Small RNA Sequencing"

_ijms, 2022, doi:10.3390/ijms231810340_

Round 1

Reviewer 1 Report

The authors of this manuscript present a study focused on the influence of anticoagulant type as a preanalytical factor on the expression profile of circulating miRNAs detected in plasma. Their study brings valuable results of quantitative changes  in expression of platelet-derived  and red blood cell-derived miRNA related to anticoagulant type in blood collection tube. Contamination of plasma by red blood cells and platelets distort the results of circulating miRNA studies and in many cases lead to inconsistent results between studies. Therefore, it is necessary to focus on the study of impact of various factors with the aim of standardizing the protocol for plasma processing and miRNA quantification in plasma.

I have a few comments:

Results

In the fig.1, 9, 10, 11 and 12, you present differences between ACD, citrate, CTAD and EDTA collection tubes in various parameters by the boxplots, but it is not stated if the differences are statistical significant. Statistical evaluation of results is missing.

In the section 2.4., miRNA levels were normalized to miR-30e-5p as reference miRNA. It should be stated if you experimentally confirmed the stable expression of this miRNA in your study sample.   

Material and methods

In the part "study sample", the statement about study approval by ethic committee and informed consent is missing.

In the part "plasma preparation", you described a workflow for platelet-free plasma preparation, but you did not measure platelet content in plasma as evidence of PFP. A study describing this centrifugation protocol and providing evidence of obtaining platelet-free plasma should by cited.

Author Response

We thank the reviewer for the insightful and useful comments. We have addressed the points raised individually below.

In the fig.1, 9, 10, 11 and 12, you present differences between ACD, citrate, CTAD and EDTA collection tubes in various parameters by the boxplots, but it is not stated if the differences are statistically significant. Statistical evaluation of results is missing.

We have added p-values of statistical significance using the appropriate statistical tests: Mann-Whitney U test for metadata and qPCR data in Fig. 1, 11, and 12, or pairwise quasi-likelihood F-test for sequencing data on Fig. 9 and 10.

In the section 2.4., miRNA levels were normalized to miR-30e-5p as reference miRNA. It should be stated if you experimentally confirmed the stable expression of this miRNA in your study sample.

The stability of miRNA expression in plasma was tested using the NormFinder algorithm on the qPCR data, and miR-30e-5p showed the lowest stability value (which means that it was the most stable miRNA). The least stable miRNA was miR-451a. We have added the description of this evaluation in the “Materials and Methods” section (p. 4.7). The NormFinder stability values have been provided in Table A3, Appendix A.

In the part "study sample", the statement about study approval by ethic committee and informed consent is missing.

The statement about study approval by ethic committee and informed consent has been added to the appropriate section. For a more detailed description, see the “Institutional Review Board Statement” and “Informed Consent Statement” sections in the manuscript.

In the part "plasma preparation", you described a workflow for platelet-free plasma preparation, but you did not measure platelet content in plasma as evidence of PFP. A study describing this centrifugation protocol and providing evidence of obtaining platelet-free plasma should be cited.

Thank you for the suggestion. We have added this information to the appropriate section.

We have also added citations for several bioinformatics tools used in the data analysis. We have checked the use of all abbreviations in the manuscript, and have corrected repeated explanations or added explanations where they are needed.

Reviewer 2 Report

Andrey V. Zhelankin and coworkers studied differences in plasma miRNA composition dependent on the anticoagulant, in healthy human volunteers. However, the also examined technical aspects across anticoagulants. Their study was conducted meticulously and reported elegantly. They report differences between anticoagulants that can be attributed to hemolysis, and differences between sequencing and qPCR that remain unexplained.

Major comment:

With regards to the concordance between the miRNA relative expression values calculated from the qPCR and sequencing data the authors should acknowledge their limitation of not applying proteinase digestion prior to organic extraction, thus allowing residual nuclease and phosphate activity after phase separation (see ref #51). Also, the presentation of concordance is flawed in that a separate scatter plot should be presented for each miRNA examined by qPCR and sequencing, wherein each data point is a single sample; and such blots should further faceted or the anticoagulant (or otherwise represent the anticoagulant).  

Minor comments:

Spell-out the acronyms for citrate-based anticoagulants

Regarding "From each participant, four plasma samples were obtained using different types of anticoagulants during blood collection" – was there a specific order for the tubes?

In Fig.2 panels A-C samples should be ordered by anticoagulant and then by subject and not vice versa.

Please explain about the Shannon diversity index.

Error bars are missing in Fig.4C.

Concerning "Polymorphic modifications (i.e., those in which the miRNA sequence had single nucleotide polymorphisms compared to the canonical miRNA sequence) were present in ~5% of the sequences" – were such SNPs detected within the seed sequence (i.e. nucleotides 2-8)?

Why was miR-30e-5p chosen as reference miRNA? To me, it seems unfit for the task.

Please verify that p-values reported in box plots such as Fig.S1 are based on a model that includes a patient blocking factor.

Author Response

We thank the reviewer for the insightful and useful comments. We have addressed the points raised individually below.

With regards to the concordance between the miRNA relative expression values calculated from the qPCR and sequencing data the authors should acknowledge their limitation of not applying proteinase digestion prior to organic extraction, thus allowing residual nuclease and phosphate activity after phase separation (see ref #51).

For miRNA isolation, we followed the protocol given in the NextPrep Magnazol cfRNA Isolation Kit (PerkinElmer) manual, which does not assume additional proteinase treatment of the plasma sample prior to organic extraction. The extraction reagent from this kit includes guanidinium, which is the most common chemical used in RNA purifications and considered the gold standard for inactivating diverse RNases [Bender et al., 2020]. Also, proteinase K is unable to fully deactivate serum RNases. It was found that high concentrations of proteinase K are unable to eliminate RNase activity in serum, unless used in concert with denaturing concentrations of SDS [Bender et al., 2020]. The proteinase digestion step could potentially have been added to the protocol independent of the kit recommendations and could possibly have reduced the risk of residual enzymatic activity. However, the effectiveness of this protocol in eliminating nucleases compared to the use of Magnazol only is not obvious.

References:

Bender, A.T.; Sullivan, B.P.; Lillis, L.; Posner, J.D. Enzymatic and Chemical-Based Methods to Inactivate Endogenous Blood Ribonucleases for Nucleic Acid Diagnostics. J Mol Diagn 2020, 22, 1030–1040, doi:10.1016/j.jmoldx.2020.04.211.

Also, the presentation of concordance is flawed in that a separate scatter plot should be presented for each miRNA examined by qPCR and sequencing, wherein each data point is a single sample; and such blots should further faceted or the anticoagulant (or otherwise represent the anticoagulant).  

We agree with the reviewer. We built separate plots comparing the relative expression measured by qPCR and sequencing for each miRNA and color-coded the study groups according to the type of anticoagulant used (Figure 13). The caption to Figure 13 has been changed accordingly.  In the text of the manuscript we also described the results based on this plot (section 2.4, lines 329-340).

Spell-out the acronyms for citrate-based anticoagulants

Corrected according to the reviewer's comment.

Regarding "From each participant, four plasma samples were obtained using different types of anticoagulants during blood collection" – was there a specific order for the tubes?

Blood sampling was performed in the following order for all samples: sodium citrate, CTAD, ACD-B, K2 EDTA. We have added this information to the section 2.1.

In Fig.2 panels A-C samples should be ordered by anticoagulant and then by subject and not vice versa.

Corrected according to the reviewer's comment.

Please explain about the Shannon diversity index.

We hypothesized that applying the alpha diversity approach to our data would help demonstrate differences in miRNA diversity between the compared groups. The Shannon index is an abundance-sensitive measure of species diversity within a community and is commonly used as an alpha diversity metric in microbial NGS analysis and population genetics.

The explanation of the Shannon diversity index and the reason for applying it to our data have been added to the manuscript, sections 2.3.1 and 4.6.

Error bars are missing in Fig.4C.

Corrected according to the reviewer's comment. We have added error bars indicating standard error of the mean.

Concerning "Polymorphic modifications (i.e., those in which the miRNA sequence had single nucleotide polymorphisms compared to the canonical miRNA sequence) were present in ~5% of the sequences" – were such SNPs detected within the seed sequence (i.e. nucleotides 2-8)?

In the QuickMIRSeq settings, alignments were allowed with one mismatch in the first 12 nucleotides of the read. The maximum number of mismatches per alignment was two. The most represented isomiR species with nucleotide variants can be found in Table S1. Among the isomiRs with polymorphic modifications in the seed region (2-8), only sporadic cases have been recorded (namely, nucleotide variance in position 8 of miR-4433b-3p, and in position 6 of miR-7977).

Why was miR-30e-5p chosen as reference miRNA? To me, it seems unfit for the task.

Reference miRNA for normalization of the qPCR data, miR-30e-5p, was chosen on the basis of the previous study which analyzed the stability of miRNAs in plasma (see ref. #54). The stability of miRNA expression in plasma was also tested using the NormFinder algorithm on our qPCR data, and miR-30e-5p had the lowest stability value (which means that it was the most stable miRNA). The least stable miRNA was miR-451a. We have added the description of this evaluation in the “Materials and Methods” section (p. 4.7). The NormFinder stability values have been provided in Table A3, Appendix A.

Please verify that p-values reported in box plots such as Fig.S1 are based on a model that includes a patient blocking factor.

All p-values of significance for pairwise quasi-likelihood F-test for sequencing data (indicated in the boxplots in Fig. 9, 10, S1, and S2) are based on a model that includes a patient blocking factor.

We have also added citations for several bioinformatics tools used in the data analysis. We have checked the use of all abbreviations in the manuscript, and have corrected repeated explanations or added explanations where they are needed.